# Combined Treatment with Two Water Extracts of *Eleutherococcus senticosus* Leaf and Rhizome of *Drynaria fortunei* Enhances Cognitive Function: A Placebo-Controlled, Randomized, Double-Blind Study in Healthy Adults

**DOI:** 10.3390/nu12020303

**Published:** 2020-01-23

**Authors:** Chihiro Tohda, Mie Matsui, Yuna Inada, Ximeng Yang, Tomoharu Kuboyama, Yoshiyuki Kimbara, Hidetoshi Watari

**Affiliations:** 1Division of Neuromedical Science, Institute of Natural Medicine, University of Toyama, Toyama 930-0194, Japan; ximeng@inm.u-toyama.ac.jp (X.Y.); kuboyama@inm.u-toyama.ac.jp (T.K.); 2Resilio Company Limited, Tokyo 107-0052, Japan; 3Laboratory of Clinical Cognitive Neuroscience, Institute of Liberal Arts and Science, Kanazawa University, Kanazawa 920-1192, Japan; miematsui@staff.kanazawa-u.ac.jp (M.M.); yunainada@outlook.com (Y.I.); 4Department of Japanese Oriental Medicine, Graduate School of Medicine and Pharmaceutical Sciences, University of Toyama, Toyama 930-0194, Japan; zfzyknbr@med.u-toyama.ac.jp (Y.K.); watari44@med.u-toyama.ac.jp (H.W.)

**Keywords:** *Eleutherococcus senticosus* leaves, *Drynaria fortunei* rhozomes, cognitive function, Healthy subject, anti-stress

## Abstract

We previously found that the water extract of *Eleutherococcus senticosus* leaves (ES extract) enhanced cognitive function in normal mice. Our study also revealed that the water extract of rhizomes of *Drynaria fortunei* (DR extract) enhanced memory function in Alzheimer’s disease model mice. In addition, our previous experiments suggested that a combined treatment of ES and DR extracts synergistically improved memory and anti-stress response in mice. Although those two botanical extracts are expected to be beneficial for neuropsychological function, no clinical data has ever been reported. Therefore, we performed a placebo-controlled, randomized, double-blind study to evaluate cognitive enhancement and anti-stress effects by the intake of a combined extract in healthy volunteers. The intake period was 12 weeks. The Japanese version of the Repeatable Battery for the Assessment of Neuropsychological Status (RBANS) test was used for neurocognitive assessment. The combined treatment of ES and DR extracts significantly increased the figure recall subscore of RBANS (*p* = 0.045) in an intergroup comparison. Potentiation of language domain ((*p* = 0.040), semantic fluency (*p* = 0.021) and figure recall (*p* = 0.052) was shown by the extracts (in intragroup comparison). In anti-stress response, the anxiety/uncertainly score was improved by the extract in an intragroup comparison (*p* = 0.022). No adverse effects were observed. The combined treatment of ES and DR extracts appear to safely enhance a part of cognitive function in healthy adults.

## 1. Introduction

In this study, we focus on two crude drug extracts, leaf of *Eleutherococcus senticosus* and rhizome of *Drynaria fortunei.* Our previous studies have indicated these two extracts potentiate cognitive function in mice, explained as below, although no human study has been performed yet. No adverse effect was reported in animal studies concerning these two extracts.

*E. senticosus* (Rupr. and Maxim.) Maxim. (synonymous with Acanthopanax senticosus), also known as “Siberian Ginseng” (English), “Ciwujia” (Chinese), or “Ezoukogi” (Japanese), is a species of woody shrub in the family Araliaceae [1]. The rhizomes and roots of *E. senticosus* are recorded in Chinese and Japanese pharmacopoeias as a treatment for neurasthenia, hypertension, chronic coughing, and ischemic heart disease. In contrast, *E. senticosus* leaf is classified as food and has been taken as tea, soup, wine, and so on. In vivo pharmacological data of the leaf extract has hardly been reported, except for reducing the activity of triglycerides in high-fat diet-fed mice [2].

We previously investigated the effect of a water extract of *E. senticosus* leaves (ES extract) on the cognitive function in normal mice and determined which active constituents passed the blood–brain barrier (BBB) [3]. Oral administration of the leaf extract significantly enhanced object recognition memory. Compounds absorbed in the blood and the brain after oral administration of the leaf extract were detected by LC-MS/MS analyses. Detected compounds in plasma and the cerebral cortex were ciwujianoside C3, eleutheroside M, ciwujianoside B, and ciwujianoside A1 [3]. Those compounds themselves significantly enhanced object recognition memory by oral administration in normal mice and extended the length of dendrites in cultured cortical neurons [3].

The dried rhizomes of *D. fortunei* (Kunze et Mett.) J. Sm., known as “Drynariae Rhizoma”, is a widely distributed traditional medicine in mainland China, Korea, and Japan [4]. It is reported to tonify the kidneys, strengthen bones, and promote the healing of fractures [5]. Flavonoids are the main constituents of *D. fortunei* rhizomes and have shown activity in animal experiments against osteoporosis, bone fractures, oxidative damage, and inflammation [6,7,8,9].

We previously found that the water extract of Drynaria Rhizome (DR extract) could enhance memory function and ameliorate Alzheimer’s disease (AD) pathologies in 5XFAD model mice [10]. A biochemical analysis led to the identification of the bio-effective metabolites that are transferred to the brain, naringenin and its glucuronides [10]. Naringenin directly bound to collapsin response mediator protein 2 protein (CRMP2). The water extract of Drynaria Rhizome and naringenin induced axonal growth in cultured cortical neurons [10].

In our previous experiments (published in patent #JP6165380), simultaneous treatment of ES extract and DR extract synergistically improved memory dysfunction in 5XFAD mice. Besides cognitive enhancement, our preliminary experiment using normal mice showed the synergic effect of ES and DR extracts on a reduction of depressive behavior in forced swim tests. Although those two botanical extracts were expected to have beneficial effects on cognitive function and anti-stress, as shown in previous animal studies, no clinical data has ever been reported. Therefore, we conducted a clinical study to evaluate cognitive enhancement and anti-stress effects by the intake of ES extract plus DR extract. Both *E. senticosus* leaves and rhizomes of *Drynaria fortunei* are classified as non-pharmaceuticals by the Pharmaceutical and Medical Device Act in Japan.

## 2. Materials and Methods 

### 2.1. Trial Design

This placebo-controlled, randomized, double-blind study in healthy adults was conducted with the approval of the Ethics Committee of the University of Toyama. Each subject signed an informed consent form prior to study entry. The potential subjects (*n* = 31) were allocated into two groups. All subjects who met the inclusion criteria were enrolled; data from 31 subjects were finally analyzed. In the case of blood tests, samples of 2 subjects in the placebo group and 3 subjects in the extract group were chylous due to a meal before sampling. Therefore, those blood data were omitted. All subjects visited the University of Toyama two times for testing. Further details of the CONSORT flowchart of the study are shown in Figure 1.

### 2.2. Participants

The period of subject recruitment was from 1 December 2018 to 4 March 2019. The inclusion criteria for eligible subjects were as follows: (a) an age of ≥40 and ≤80 years; (b) volunteers to attend this clinical study; (c) good physical and mental health. The exclusion criteria were as follows: (a) under 39 years old; (b) pregnant or lactating; (c) subjects having mental illnesses; (d) subjects judged being inappropriate for other reasons. Subjects were followed up from 13 April 2019 to 17 August 2019.

### 2.3. Intervention 

ES extract was prepared by TOKIWA Phytochemical Co., Ltd (Chiba, Japan) as follows. Fresh leaves of *E. senticosus* were purchased from China. The dried powder of *E. senticosus* leaves (4 kg) was extracted in hot water (85 °C, 40 L) for 30 min, with this stage repeated twice. The liquid portion was then combined, filtered, and lyophilized to yield a leaf water extract (yield 30.0%). Contents of active principles in the ES extract, ciwujianoside C3, eleutheroside M, ciwujianoside B, were 1.33%, 6.29%, 0.95%, respectively. Safety assessments of the ES extract were performed by Japan Food Research Laboratories and BoZo Research Center. Results showed no acute toxicity in mice (LD50 was more than 2000 mg/kg) and no gene mutagenesis.

DR extract, the water extract of *D. fortunei*, was purchased from BGG Japan Co., Ltd. (Tokyo, Japan). The content of naringin in the extract was 22.3%. Naringin is metabolized to naringenin in the body after oral administration. Safety assessments of the DR extract were performed by College of Applied Arts and Science of Beijing Union University. Results showed no acute toxicity in mice (LD50 was more than 5000 mg/kg), no chronic toxicity (after 30 days administration), and no gene mutagenesis.

One capsule contains 67.67 mg of ES extract, 6.67 mg of DR extract, 44.46 mg crystal cellulose, and 1.20 mg calcium stearate. Three capsules, once per day, were taken. Placebo capsules contain 118.8 mg crystal cellulose and 1.20 mg calcium stearate. Three capsules, once per day, were taken. Making capsules and packaging was done by the manufacturer (Aliment Industry Co., Ltd., Yamanashi, Japan) under Good Manufacturing Practice controls and ISO22000 certification. Intake dose per day was 203 mg of ES extract plus 20 mg of DR extract. Effective minimum doses and best-combination of doses were determined by our previous animal studies.

### 2.4. Outcomes and Assessments

All participants completed a basic sociodemographic and medical history questionnaire and reported any medications used at baseline. The Japanese version of the Repeatable Battery for the Assessment of Neuropsychological Status (RBANS) was performed as the primary neurocognitive outcome measure. For measuring stress responses, the public health research foundation stress checklist short form (PHRF-SCL) was used [11]. This test evaluates 4 factors: autonomic symptoms, tiredness/physical responses, anxiety/uncertainty, depression/feeling of insufficiency. The Japanese version of the Mini Mental State Examination (MMSE) was administered as a secondary outcome measure.

### 2.5. Neurocognitive Assessments

The RBANS, a representative, clinician-administered neuropsychological test for adults, was used to assess multiple cognitive function domains [12]. This test includes 12 standard cognitive subtests. The results are grouped into the following five domains: immediate memory (list learning and story memory), visuospatial/constructional (figure copy and line orientation), language (picture naming and semantic fluency), attention (digit span and digit symbol coding), and delayed memory (list recall, list recognition, story recall, and figure recall). The reliability and validity of the Japanese version of the RBANS has been well-established [13], and at least two forms were prepared to avoid the effect of learning via test repetition. As noted above, the MMSE [14] was also applied. The Japanese Adult Reading Test (JART) was used to estimate the intelligence quotients (IQs) of the subjects as a background measure.

### 2.6. Safety Assessment

The safety assessment included the recording of adverse events and the conducting of biochemical blood tests to assess liver and renal function and blood sugar and lipid levels at each visit.

### 2.7. Randomization

Participants were randomly assigned to one of two groups, the extract group or the placebo group. Randomization was performed by a block randomization method by a third-party (TOKIWA Phytochemical Co., Ltd.) who secured the participant allocation list and performed a key opening.

### 2.8. Statistical Analysis

The results are expressed as means with standard deviations (SD). A 95% confidence interval (CI) is the interval that is 95% certain to contain the true population value, as it might be estimated from a much larger study. 95% CI limits are shown by lower and upper ranges. Statistical comparisons were performed using GraphPad Prism 6 (GraphPad Software, La Jolla, CA, USA). Data were analyzed using two-tailed unpaired *t*-tests (for intergroup comparison) or two-tailed paired *t*-test (for intragroup comparison) and *p* values <0.05 were considered significant. 

## 3. Results

### 3.1. Baseline Characteristics of the Study Groups

The overall study population was comprised of 31 healthy men and women. Figure 1 presents the CONSORT study flow diagram, subject distribution, and individual study protocols. Thirty-one candidates were randomized into two groups. One group was allocated to a 12-week placebo group, and another one was allocated to a 12-week extract group. All subjects completed all tests and were analyzed, and their characteristics are shown in Table 1. The analyses of cognition and stress were done based on the full analysis set. Education periods and MMSE scores immediately before the test in extract and placebo groups were not significantly different. Assumed IQs were also not significantly different between groups. Since the average IQ in Japanese is around 100, subjects in this study have a normal level of intelligence. 

### 3.2. Neuropsychological Functioning

The main analyses of RBANS compared the total score, 5 domain scores, and 12 subdomain scores between the extract and placebo groups (Table 2). When compared between those groups, a subdomain of “figure recall” was significantly increased by extract intake (*p* = 0.045). When compared intragroup at pre and post administrations, the extract treatment significantly increased the “language” domain (*p* = 0.040) and subdomains of “semantic fluency” (*p* = 0.021) and “figure recall” (*p* = 0.052).

MMSE scores were not significantly changed between groups and also pre–post administration because participating subjects had already been marked with nearly full scores at the initial test (Table 3). Stress responses were evaluated by PHRF-SCL, which is one of the established self-report scales that detect both physical and psychological stress (Table 4). Although an intergroup comparison showed no significant changes in all items, the anxiety/uncertainty item was significantly reduced by extract intake when compared pre and post administrations (*p* = 0.022). Depression/feeling of insufficiency and total score tended to be reduced by extract intake in the intragroup comparison.

### 3.3. Safety Measures

Thirty-one parameters were analyzed in blood to evaluate the incidence of side effects. Adverse effects were investigated following both extract and placebo intakes. Table 5 lists the results of the safety evaluation. Interestingly, total cholesterol level and HDL-cholesterol level were significantly decreased with placebo intake. Other parameters showed no significant differences between groups.

## 4. Discussion

Intake of a combination of ES extract and DR extract for 12 weeks gave significant enhancement to the figure recall function in the RBANS test (Table 2). In an intragroup comparison of pre- and post-intake, language domains were also potentiated by the combined extract treatment (Table 2). Among stress responses, anxiety/uncertainty was ameliorated by the extract intake, but not by the placebo intake. (Table 4). The extract treatment showed no adverse effects (Table 5). Effects of ES extract or DR extract on the cognitive function and stress response in humans have not been reported yet. This study is the first to show the potential of a combined extract of ES and DR against neuropsychological function. In our previous studies, ciwujianoside C3, eleutheroside M and ciwujianoside B were identified as transferred compounds in the brain of mice after the intake of ES extract [3]. No pharmacological activities of those compounds have not reported except for our previous report. Ciwujianoside C3, eleutheroside M, and ciwujianoside B enhanced memory function in normal mice and increased dendrite densities of cultures cortical neurons [3]. Those effects might also be related to cognitive enhancement in humans, as shown in this study. For unraveling the signaling pathways of ciwujianoside C3, eleutheroside M, and ciwujianoside B involved in cognitive function, we are now identifying direct binding molecules of ciwujianoside C3, eleutheroside M, and ciwujianoside B using the DARTS method [10].

Our previous study indicated that only naringenin and its glucuronides penetrated the brain after oral administration of DR extract [10]. Naringenin itself enhanced memory function and axonal density in Alzheimer’s model mice [10]. Naringenin directly binds to CRMP2 and ameliorates abnormal phosphorylation of CRMP2 in neurons, resulting in axonal growth [10]. Efficacy of DR extract in humans, shown in this study, might be due to CRMP2 regulation. Other groups have reported that naringenin treatment attenuates anxiety-like behavior in socially defeat stressed mice [15]. Taken together, ES extract and DR extract may cooperate against neuropsychological function with different signal pathways. In our other study, diosgenin-rich yam extract that showed axonal growth activity and memory enhancement effects in mice facilitated human cognitive function [16], indicating neurite extension activity might be related to the upregulation of cognition.

Several limitations of this investigation should be noted. We enrolled a 43–79 years sample of Asian adults, and therefore our results might not be generalizable to other populations. Furthermore, we did not assess daily dietary intake and physical activity level, and we are therefore unsure of the effects of these factors on the results of this study. The participants were advised not to modify their eating habits and activity patterns during the intervention. Finally, the study is also limited by the small sample size. 

## 5. Conclusions

In conclusion, this placebo-controlled, randomized, double-blind study revealed that a combined treatment with two water extracts of Eleutherococcus senticosus leaf and rhizome of Drynaria fortunei enhances cognitive function in healthy human adults without any adverse effects.

## 6. Patents

This work is related to patent JP6165380.

## Figures and Tables

**Figure 1 nutrients-12-00303-f001:**
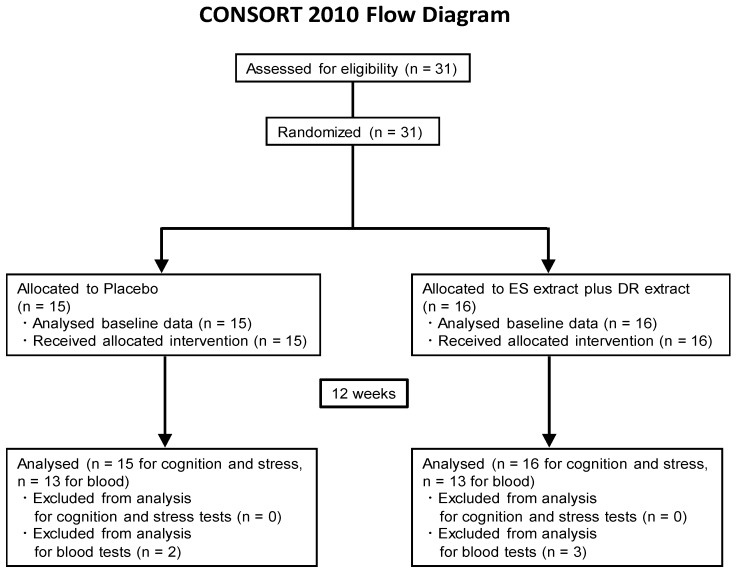
Study flow (CONSORT 2010 diagram). CONSORT, Consolidated Standards of Reporting Trials.

**Table 1 nutrients-12-00303-t001:** Sociodemographic and baseline characteristics of the sample.

	Age			Sex	Education Period			MMSE			IQ		
	Mean	SD	*p*-Value Intergroup		Mean	SD	*p*-Value Intergroup	Mean	SD	*p*-Value Intergroup	Mean	SD	*p*-Value Intergroup
**Extract**	63.0	10.63		Male, 6; female, 10	13.8	2.21		28.69	1.85		105.1	11.44	
			0.7359				0.7447			0.6593			0.9596
**Placebo**	64.3	10.05		Male, 5; female, 10	13.5	2.59		28.93	1.10		104.9	9.25	

**Table 2 nutrients-12-00303-t002:** Changes in Repeatable Battery for the Assessment of Neuropsychological Status (RBANS) scores between ES extract plus DR extract intake and placebo intake.

		Pre		Post		Intragroup Comparison (Pre vs. Post)			Changed Value (Post—Pre)		Comparison between Group (Extract vs. Placebo)		
Cognitive Domain		Mean	SD	Mean	SD	95% CI		*p*-Value	Mean	SD	95% CI		*p*-Value
						Lower	Upper				Lower	Upper	
**Total Score**	Extract	110.07	12.50	111.00	14.49	−5.597	3.730	0.674	0.93	8.42	−9.608	4.037	0.409
	Placebo	108.50	16.50	110.13	19.36	−8.104	4.854	0.601	1.63	12.16			
Immediate Memory	Extract	107.13	10.26	101.80	11.55	−0.286	10.952	0.061	−5.33	10.15	−4.220	8.078	0.525
	Placebo	106.13	12.23	104.50	14.14	−3.285	6.535	0.491	−1.63	9.22			
Visuospatial/Constructional	Extract	100.93	27.27	104.93	16.14	−19.054	11.054	0.578	−2.67	14.34	−11.784	7.213	0.625
	Placebo	105.94	17.67	102.31	16.32	−1.028	8.278	0.118	−3.63	8.73			
Language	Extract	103.60	17.97	113.33	15.73	−18.9939	−0.527	0.040 *	9.73	16.62	−22.900	11.186	0.486
	Placebo	103.38	16.91	111.75	24.76	−23.107	6.357	0.244	6.88	25.95			
Attention	Extract	106.20	11.60	109.13	9.71	−6.967	1.101	0.141	2.93	7.28	−10.288	4.860	0.468
	Placebo	108.44	12.82	110.19	16.14	−7.802	4.302	0.547	1.75	11.36			
Delayed Memory	Extract	108.60	13.63	107.13	9.66	−5.808	8.741	0.672	−1.47	13.14	−11.911	7.768	0.669
	Placebo	105.06	15.19	104.19	15.90	−7.876	9.626	0.834	−0.88	16.42			
List Learning	Extract	11.07	2.74	10.60	2.69	−1.112	2.045	0.536	−0.47	2.85	−1.236	2.522	0.488
	Placebo	10.69	2.52	10.81	2.64	−1.443	1.193	0.843	0.13	2.47			
Story Memory	Extract	11.47	2.33	10.13	1.92	0.315	2.352	0.014 *	−1.33	1.84	−2.359	1.787	0.779
	Placebo	11.19	2.74	10.19	3.43	−0.783	2.783	0.251	−1.00	3.35			
Figure Copy	Extract	10.67	1.05	10.20	1.90	−0.577	1.510	0.354	−0.47	1.88	−1.314	1.314	1.000
	Placebo	10.88	2.03	10.19	2.01	−0.108	1.483	0.085	−0.69	1.49			
Line Orientation	Extract	10.40	2.29	10.47	2.53	−1.524	1.391	0.923	0.07	2.63	−2.397	1.683	0.722
	Placebo	9.94	2.74	10.06	3.53	−1.513	1.263	0.850	0.13	2.60			
Picture Naming	Extract	9.67	1.68	10.13	1.77	−1.884	0.951	0.492	0.47	2.56	−3.772	1.343	0.338
	Placebo	9.63	2.22	9.38	2.99	−1.859	2.359	0.804	−0.25	3.96			
Semantic Fluency	Extract	10.33	3.72	12.67	2.79	−4.259	−0.407	0.021 *	2.33	3.48	−3.466	2.609	0.774
	Placebo	10.69	1.96	12.75	3.66	−4.207	0.082	0.058	2.06	4.02			
Digit Span	Extract	9.07	2.37	10.00	1.96	−2.107	0.241	0.110	0.93	2.12	−2.405	0.977	0.393
	Placebo	9.88	2.73	10.38	3.03	−1.667	0.667	0.376	0.50	2.19			
Digit Symbol Coding	Extract	12.27	3.13	12.80	2.78	−1.821	0.755	0.389	0.53	2.33	−2.154	1.582	0.756
	Placebo	11.88	3.28	12.38	3.56	−1.746	0.746	0.406	0.50	2.34			
List Recall	Extract	11.27	2.22	11.20	2.70	−1.360	1.493	0.922	−0.07	2.58	−1.754	2.326	0.776
	Placebo	10.56	2.61	10.69	2.18	−1.579	1.329	0.857	0.13	2.73			
List Recognition	Extract	10.27	1.62	9.93	2.19	−0.863	1.530	0.560	−0.33	2.16	−2.190	1.332	0.621
	Placebo	10.13	2.06	9.75	2.54	−0.999	1.749	0.569	−0.38	2.58			
Story Recall	Extract	11.47	2.59	10.93	1.79	−0.930	1.997	0.447	−0.53	2.64	−2.095	1.666	0.817
	Placebo	11.00	2.66	10.38	2.36	−0.421	1.671	0.222	−0.63	1.96			
Figure Recall	Extract	11.33	2.29	12.40	2.61	−2.143	0.010	0.052 *	1.07	1.94	−3.250	−0.036	0.045 *
	Placebo	11.88	2.31	11.06	3.30	−0.292	1.917	0.138	−0.81	2.07			

SD, standard deviation; **p* < 0.05.

**Table 3 nutrients-12-00303-t003:** Changes in the Mini Mental State Examination (MMSE) between ES extract plus DR extract intake and placebo intake.

		Pre		Post		Intragroup Comparison (Pre vs. Post)			Changed Value (Post—Pre)		Comparison between Group (Extract vs. Placebo)		
		Mean	SD	Mean	SD	95% CI		*p*-Value	Mean	SD	95% CI	*p*-Value	
						Lower	Upper				Lower	Upper	
**MMSE**	Extract	28.93	1.10	29.20	1.21	−0.7989	0.2656	0.301	0.27	0.96	−0.976	1.118	0.890
	Placebo	28.69	1.85	29.00	1.15	−1.1315	0.5065	0.429	0.31	1.54			

**Table 4 nutrients-12-00303-t004:** Changes in stress responses between ES extract plus DR extract intake and placebo intake.

		Pre		Post		Intragroup Comparison (Pre vs. Post)			Changed Value (Post–Pre)		Comparison between Group (Extract vs. Placebo)		
		Mean	SD	Mean	SD	95% CI		*p*-Value	Mean	SD	95% CI		*p* Value
						Lower	Upper				Lower	Upper	
**Total Score**	Extract	11.20	6.99	9.20	7.00	−0.414	4.414	0.097	−2.00	4.36	−3.946	4.661	0.866
	Placebo	12.38	8.85	10.38	6.82	−1.296	5.296	0.215	−2.00	6.19			
Autonomic Symptoms	Extract	0.87	1.13	0.93	1.49	−0.744	0.610	0.836	0.07	1.22	−1.252	.252	0.183
	Placebo	1.44	1.15	0.94	1.06	0.163	0.837	0.006 **	−0.50	0.63			
Tiredness/Physical Responses	Extract	4.33	2.82	3.93	3.03	−0.483	1.283	0.348	−0.40	1.59	−1.226	2.226	0.557
	Placebo	4.94	3.51	4.88	3.74	−1.306	1.431	0.924	−0.06	2.57			
Anxiety/Uncertainly	Extract	3.00	2.33	2.13	2.20	0.146	1.588	0.022 *	−0.87	1.30	−1.435	1.435	1.000
	Placebo	2.69	2.44	1.81	1.94	−0.290	2.040	0.130	−0.88	2.19			
Depression/Feeling of Insufficiency	Extract	3.00	2.51	2.20	2.04	−0.117	1.717	0.082	−0.80	1.66	−1.054	1.768	0.607
	Placebo	3.31	2.98	2.75	2.02	−0.448	1.573	0.254	−0.56	1.90			

SD, standard deviation; **p* < 0.05, ***p* < 0.01.

**Table 5 nutrients-12-00303-t005:** Changes in blood data between ES extract plus DR extract intake and placebo intake.

	Changed Value					
	Extract		Placebo			
	Mean	SD	Mean	SD	*p*-Value	CI (95%)
Total Cholesterol	0.46	15.86	−15.23	21.29	0.044 *	0.4956 to 30.89
HDL-Cholesterol	−3.15	3.98	−9.62	5.32	0.002 *	2.662 to 10.260
LDL-Cholesterol	2.85	14.84	−5.15	14.60	0.179	−3.918 to 19.920
Albumin	−0.08	0.13	−0.25	0.28	0.070	−0.014 to 0.337
Total Protein	−0.17	0.20	−0.32	0.42	0.269	−0.121 to 0.413
BIL Direct	0.02	0.04	0.01	0.06	0.712	−0.035 to 0.050
BIL Indirect	0.02	0.13	0.05	0.16	0.588	−0.146 to 0.085
Glucose	−1.23	13.75	−2.69	26.21	0.860	−15.480 to 18.400
TG	13.69	78.08	1.92	43.42	0.639	−39.37 to 62.91
Urea Nitrogen	0.03	1.86	1.06	3.05	0.308	−3.074 to 1.013
Creatinine	0.02	0.04	0.01	0.03	0.469	−0.018 to 0.038
Uric Acid	0.02	0.64	0.14	0.53	0.621	−0.591 to 0.361
Na	0.23	1.83	0.38	1.66	0.824	−1.569 to 1.262
K	0.05	0.30	0.17	0.32	0.356	−0.368 to 0.137
Cl	0.77	1.83	1.54	1.81	0.292	−2.243 to 0.705
Amylase	−8.54	11.30	−3.85	8.51	0.244	−12.790 to 3.408
CK	10.92	49.41	−26.23	48.09	0.064	−2.316 to 76.620
LAP	−0.15	3.78	3.23	17.71	0.507	−13.750 to 6.980
g-GTP	1.85	3.91	9.85	32.22	0.383	−26.580 to 10.580
Cholinesterase	−14.31	16.77	−24.15	32.14	0.337	−10.910 to 30.600
AST(GOT)	−0.31	2.50	0.46	6.97	0.711	−5.008 to 3.470
ALT(GPT)	−1.77	3.06	0.69	9.84	0.398	−8.362 to 3.439
LD(LDH)	−13.46	18.91	−28.00	23.29	0.093	−2.634 to 31.710
ALP	−22.31	29.45	5.15	71.10	0.211	−71.520 to 16.590

SD, standard deviation; **p* < 0.05.

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
