# Peer review of "Combined Treatment with Two Water Extracts of Eleutherococcus senticosus Leaf and Rhizome of Drynaria fortunei Enhances Cognitive Function: A Placebo-Controlled, Randomized, Double-Blind Study in Healthy Adults"

_nutrients, 2020, doi:10.3390/nu12020303_

Round 1

Reviewer 1 Report

The work presented in this manuscript explores the possibility of enhancing cognitive functions in human volunteers by providing water extracts of Eleutherococcus senticosus leaves and that of rhizomes of Drynaria fortunei .The duration of the experiment was for 12 weeks. The work is an extension of their earlier work on mice model. The placebo and experimental groups were assessed for neuropsychological status. The results indicated a significant increase in the sub scores of tests assessed.Anti stress response was also improved by the intake of water extracts of two raw materials used.

Though the results are encouraging I find several weak points in the experimental design which authors themselves have identified. This include very low number of human subjects used in the study,large differences in the age group of subjects, no identification of active principles from extracts which could have been used as markers for repeated experiments. I also noticed that there is no mention about basal cognitive status of subjects to indicate whether subjects had normal or below par cognitive status.

Author Response

1) Thank you for very suggestive comments. I agree with the limitation of this study by low number and large differences of subjects. However, shown in Table 1, mean and SD values of age in extract and placebo groups are similar with no statistical differences. Therefore, data shown in this study should be acceptable under appropriate statistical analyses.

2) Contents of active principles in ES extract and DR extract have been added in lines 100 – 102.

3) Assumed IQs were not also significantly different between groups. Since averaged IQ in Japanese is around 100, subjects in this study have the normal level of intelligence. This mention was added in lines 164 – 166.

Reviewer 2 Report

Title: Combined treatment with two water extracts of Eleutherococcus senticosus leaf and rhizome of Drynaria fortunei enhances cognitive function: a placebo-controlled, randomized, double-blind study in healthy adults

Comments:

Abstract- please mention the main results (at least the p- values) in the abstract.

Introduction- The study is about human subjects. But the introduction is missing background information on the test compounds on humans. If it the first time being tested on humans, it should be mentioned and highlighted. There needs to be a clear justification for transitioning from animal studies to human trials beyond this one line “Although those two botanical extracts are expected to have beneficial effects on cognitive function and anti-stress from previous animal studies, no clinical data has never been reported.” Please also mention some basic chemical properties with side-effects (or lack thereof) for readers who don’t know about the test compounds chosen.

Materials and methods- please revise and re-write section 2.1 accordingly. It does not match with Figure 1. The blood data that were omitted is not reflected on Fig 1.

Please provide citation for RBANS at the first mention, Line 106. If possible, please provide short details of each of the 12 components as an Appendix.

Please provide some details of each test compound in an appendix, for the lay audience: source, function, natural/synthetic, health benefits etc.

Please provide rationale for safety assessments (going back to the safety issues mentioned above).

If other studies have published methods for calculating “GM-BHQ and FA-BHQ” as used in the study, please cite appropriately. If this is a novel technique, it is important to provide the details in a supplement, for the purpose of replicability. The techniques as described currently, are unclear “GM-BHQ and FA-BHQ were calculated using the T1-weighted, T2-weighted, diffusion 105 tensor, and resting-state functional MRI images, approved using ITU-T H.861”.

The randomization procedure is not clear. Please provide accurate details so it is replicable.

Please include mean and SD in the statistical analyses section as well as calculation of delta.

Overall: Very interesting research!

Author Response

1) Thank you for very suggestive comments. We changed description in Abstract with P values shown in lines 29 – 32, and Results in lines 176 – 184.

2) Thank you for very suggestive comment. We explained in Introduction as “In this study, we focused on two crude drug extracts, leaf of Eleutherococcus senticosus and rhizome of Drynaria fortunei. Our previous studies indicated those two extracts potentiated cognitive function in mice explained as below, and no human study has been performed yet. No adverse effect was reported in animal studies concerning those two extracts. “ (lines 38 – 41).

3) We have corrected Figure 1. Description in section 2.1. is correct.

4) Thank you for comments. We already cited RBANS as reference #12 (line 130). Detail domains and subscores are shown in Table 2.

5) Extracts used in this study were extracts of natural crude drug, not synthesized compounds. Sources and general functions are already described in lines 42 – 48 for ES, line 57 – 61 for DR.

6) General safety was mentioned in Introduction (line 40- 41). Results of safety assessment tests were added in Materials and Methods in lines 102 – 104 for ES extract, and lines 107 – 109 for DR extract.

7) We can’t understand about your comments of GM-BHQ and FA-BHQ. Such kinds of contents are not included in our manuscript. It seems to be comments for other’s papers ?

8) Randomization procedure has been added in lines 146 – 147.

9) The results are expressed as means with standard deviations (SD). This was already mentioned in line 150.

Reviewer 3 Report

In the current report Tohda and collaborators, indicate that two water extracts of Eleutherococcus senticosus leaf and rhizome of  Drynaria fortunei enhance cognitive function according to the results from a  placebo-controlled, randomized, double-blind study  in healthy adults. The design of the current study is correct and results are interesting and similar to another recent study performed from the same authors regarding the enhancement of cognitive function in healthy adults following treatment with Diosgenin-Rich extracts. However, some parts of the manuscript need to be addressed.

Introduction

Introduction is generally well written although is relatively short; however, a small paragraph should be devoted to the description of molecular mechanisms of both water extracts activity, that have been used against some pathological conditions, as indicated by the authors. Moreover, authors could indicate some other examples of natural derived compounds with similar abilities in terms of cognitive improvement in healthy adults.

In addition, authors indicate that according to a recent patent (#JP6165380) simultaneous treatment of ES and DR extracts synergistically improved memory dysfunction in 5XFAD mice. However a reference is missing and should be added.

Materials and Methods

2.2 Participants

Line 88: …being inappropriate by other reasons: Which ones? Please state them

2.3 Intervention

Lines 101-102: Was the intake dose determined in both cases by animal studies, or that was only the case for DR extracts? Please provide more detailed informations

2.4/2.5/2.6

In all three sections, text is quite similar with the respective sections of a recent publication from the same group ‘’Diosgenin-Rich Yam Extract Enhances Cognitive Function: A Placebo-Controlled, Randomized, Double-Blind, Crossover Study of Healthy Adults.’’. I recommend that authors should re-phrase the respective sections.

2.7. Randomization

Line 128: …by a third party: Which one? Please state

3.Results

Lines 136-138 should be removed

3.1 Baseline characteristics of the study design

Although it is indicated that sociodemographic and baseline characteristics included IQ values in the Materials and Methods section, they are missing in Table 1 and should be added.

3.2 Neurophysiological functioning

Authors should indicate at least in the first appearance in the manuscript the meaning of the ‘’95% CI’’ value present in different tables as well as the difference between comparison of  pre and post treatments and that of intragroup comparison, in order to be more comprehensive to the readers.

Moreover in Table 2, according to the respective values ‘’language’’ was increased significantly but an asterisk is missing in the ‘’Intragroup comparison’’ and should be added

In addition in Lines 162-163, it is indicated that intergroup comparison showed no significant changes in all items, but in Table 4 there is an asterisk with value of 0.022. Was that a mistake?

3.3 Safety measures

Line 177: According to the respective values, HDL-cholesterol was also significantly reduced and should be added.

Author Response

1) About active compounds acting in the brain and their molecular mechanisms are already written in Introduction and Discussion (lines 549 – 56, 62 – 67, 246 – 257). In addition, other similar our study has been mentioned (lines 2630 – 262) and cited as reference #16.

2) Thank you for very important comments. We already cited the patent information in lines 275 -276, because it’s not like a paper, but searchable in a patent platform by the registry number.

3) In 2.2 Participants, “ (d) subjects judged being inappropriate by other reasons” has been deleted, because no subject was judged as like this.

4) Effective minimum doses and best combination of doses had been determined by our animal studies. This has been added in lines 115 – 116.

5) We have tried to re-phrase sections 2.4/2.5/2.6. However, those parts are just definitions and simple explanations of used tests. Therefore, it is difficult to re-word. Since we used identical tests previous we used, so similar explanations are not avoided. 

6) Thank you for very important comments. Randomization procedure has been added in lines 146 – 147.

7) Previous unnecessary sentences line 136 – 138 has been removed.

8) IQs have been added in Table 1. Assumed IQs were not also significantly different between groups. Since averaged IQ in Japanese is around 100, subjects in this study have normal level of intelligence. This mention was added in lines 164 – 166.

9) 95% CI has been explained in line 150 – 152.

10) We confirmed asterisks in Table 4.

11) Thank you for your correct comment. We added a description of HDL-cholesterol in line 216.

Round 2

Reviewer 1 Report

This study indicates that the water extracts of Eleutherococcus senticosus leaves and of Drynaria fortunei rhizomes can enhance the cognitive functions of human volunteers when consumed for a period of 12 weeks. The authors also tried to clarify few points which I raised earlier.

Author Response

Thank you for understanding.

Reviewer 2 Report

Please correct Figure 1. If you exclude 2 and 3 subjects respectively from N=15 and N=16, the final number analyzed should be N=13 and N=13, not 15 and 16 in the last two boxes of the diagram. Please consult a senior scientist if this point is still unclear.

Author Response

Thank you for a comment. We have clearly added correct subject numbers in new figure 1. Please conform it. 

Reviewer 3 Report

The revised version of the present work by Tohda and collaborators, has been significantly improved, as several aspects throughout the manuscript have been clarified. No more comments from my part

Author Response

Thank you for understanding.